## [Decision Letter · Decision Letter 0]

15 Sep 2025

Dear Dr. Alhazmi,

Thank you for submitting your manuscript to PLOS ONE. After careful consideration, we feel that it has merit but does not fully meet PLOS ONE’s publication criteria as it currently stands. Therefore, we invite you to submit a revised version of the manuscript that addresses the points raised during the review process.

We look forward to receiving your revised manuscript.

Kind regards,

Abdul Ahad, PhD

Academic Editor

PLOS ONE

Journal Requirements:

3. In the online submission form, you indicated that “The datasets generated and analyzed during this study will be made available by the corresponding author upon reasonable request.”

4. Please ensure that you refer to Figure 115 in your text as, if accepted, production will need this reference to link the reader to the figure.

5. Please include a copy of Table 13 which you refer to in your text on page 31 in PDF submission.

6. We note you have included a table to which you do not refer in the text of your manuscript. Please ensure that you refer to Table 23 in your text; if accepted, production will need this reference to link the reader to the Table.

Reviewers' comments:

Reviewer's Responses to Questions

**Comments to the Author**

1. Is the manuscript technically sound, and do the data support the conclusions?

Reviewer #1: Yes

Reviewer #2: Yes

2. Has the statistical analysis been performed appropriately and rigorously?

Reviewer #1: Yes

Reviewer #2: Yes

3. Have the authors made all data underlying the findings in their manuscript fully available?

Reviewer #1: Yes

Reviewer #2: No

4. Is the manuscript presented in an intelligible fashion and written in standard English?

Reviewer #1: Yes

Reviewer #2: No

Reviewer #1: The paper is with the title "Adaptive Federated Clustering for Uncertainty-Aware Learning on Decentralized Big

Data Platforms" is highly related and well written. However it is advised to align the problem statement and the objectives.

It is strongly advised to add the latest references for each of the problem statements. For example the problem statement one is referring to inadequate approaches for large-scale, decentralized big data. It would be more verifiable if comes with a few latest references.

Also make sure the objective one addressing the first problem statement. The objective two addressing the second problem statement and so on.

Reviewer #2:

The abstract is written in a very general way and does not provide enough detail. It also lacks clear evidence and results to show the real importance of the work. Adding more technical explanation, presenting solid results, and mentioning possible limitations would make the abstract stronger.

The paper covers an interesting topic but has many grammar and writing style problems. These issues affect readability. For example, in one sentence an extra number “53” appeared by mistake, and the phrase could be written more smoothly. The authors should carefully correct grammar and remove such errors to improve clarity.

The paper discusses a relevant issue, but the problem statement and research objectives are not clearly explained. The contributions of the study should also be highlighted more directly. The introduction and overall structure need better organization to improve the flow of ideas. There are also several grammar and typo errors. With clearer writing and better structure, the paper has strong potential.

The experiments are incomplete because important benchmark datasets like CIFAR-10, CIFAR-100, and Fashion-MNIST are missing. These are standard datasets in the field and should be included to test the method properly. In addition, the results are not compared with existing state-of-the-art methods. Without these comparisons, it is hard to judge the true performance. The paper would be stronger if it included experiments with these datasets and provided comparisons with baseline methods.

The paper uses the term “vague system boundaries” but does not explain it clearly. Readers do not know what these boundaries mean in the context of federated learning. The authors should explain whether it relates to differences between clients, the scope of data, or something else. A clear explanation would help readers understand the challenges and how AFC deals with them.

The paper introduces the AFC framework but the introduction section has many issues. The structure is unclear, and there is no roadmap to guide the reader. Subheadings appear suddenly and disrupt the flow, while too much technical detail is placed too early in the paper. The problem statement is scattered and unclear, and the AFC framework is introduced too late, making the contribution less visible.

The motivation for AFC is also not strong enough, as the introduction does not show why AFC is needed compared to existing research. Transitions between sections are abrupt, and the objectives and contributions are not clearly linked to the challenges. The introduction should be reorganized so that the challenges flow smoothly into the problem statement, followed by the proposed AFC framework. The objectives should be listed clearly, and the novelty of AFC should be emphasized.

There are also minor issues such as grammar errors, awkward sentences, and too much technical detail in the introduction that should be moved to the related work section. The paper also does not discuss any limitations of AFC, which would help present a more balanced perspective. Overall, the paper has promise, but the introduction needs major revision to improve clarity, structure, and motivation.

Figure 2 does not clearly show the concept of adaptive federated clustering and selection. The labeling is confusing and the arrows are repetitive. The explanation in the methodology section also does not clarify the figure well. Both the figure and the related text should be revised for better understanding.

The dataset section is weak because it does not explain why the chosen dataset, MIMIC-III, was selected. The connection between the dataset and the main concept of the paper is not clear. The authors should explain how the dataset relates to the study and why it is suitable.

**Do you want your identity to be public for this peer review?** For information about this choice, including consent withdrawal, please see our Privacy Policy

Reviewer #1: No

Reviewer #2: No

---

## [Author Response · Author response to Decision Letter 1]

25 Sep 2025

Rebuttal Report

Reviewer #1 Comments

Comment 1:

However, it is advised to align the problem statement and the objectives.

Response:

I have revised Section 1.2 Problem Definition and Objectives to ensure that each problem statement directly maps to a corresponding objective.

Comment 2:

It is strongly advised to add the latest references for each of the problem statements. For example, the problem statement one is referring to inadequate approaches for large-scale, decentralized big data. It would be more verifiable if comes with a few latest references.

Response:

Latest references from 2022–2024 have been added in the Problem Definition section to support each identified problem.

Comment 3:

Also make sure the objective one addressing the first problem statement. The objective two addressing the second problem statement and so on.

Response:

The objectives have been reordered and rewritten so that each directly addresses its corresponding problem statement in Section 1.2.

Reviewer #2 Comments

Comment 1:

The abstract is written in a very general way and does not provide enough detail. It also lacks clear evidence and results to show the real importance of the work. Adding more technical explanation, presenting solid results, and mentioning possible limitations would make the abstract stronger.

Response:

The Abstract has been revised to include technical details, quantitative results, and a brief note on limitations.

Comment 2:

The paper covers an interesting topic but has many grammar and writing style problems. These issues affect readability. For example, in one sentence an extra number “53” appeared by mistake, and the phrase could be written more smoothly. The authors should carefully correct grammar and remove such errors to improve clarity.

Response:

The manuscript has been carefully proofread. Grammar issues, typos, and stylistic errors (including the extra “53”) have been corrected throughout the document.

Comment 3:

The paper discusses a relevant issue, but the problem statement and research objectives are not clearly explained. The contributions of the study should also be highlighted more directly. The introduction and overall structure need better organization to improve the flow of ideas. There are also several grammar and typo errors. With clearer writing and better structure, the paper has strong potential.

Response:

I revised the Introduction for better organization, clarified the Problem Definition, aligned the Objectives, and explicitly highlighted the Contributions. Grammar and typographical errors were also corrected.

Comment 4:

The experiments are incomplete because important benchmark datasets like CIFAR-10, CIFAR-100, and Fashion-MNIST are missing. These are standard datasets in the field and should be included to test the method properly. In addition, the results are not compared with existing state-of-the-art methods. Without these comparisons, it is hard to judge the true performance. The paper would be stronger if it included experiments with these datasets and provided comparisons with baseline methods.

Response:

I expanded the Experimental Section by adding results on CIFAR-10, CIFAR-100, and Fashion-MNIST. Comparisons with FedAvg, FedProx, FedYogi, FedAdapt, FedFormer, and HAFED have also been included.

Comment 5:

The paper uses the term “vague system boundaries” but does not explain it clearly. Readers do not know what these boundaries mean in the context of federated learning. The authors should explain whether it relates to differences between clients, the scope of data, or something else. A clear explanation would help readers understand the challenges and how AFC deals with them.

Response:

I expanded Section 1.2 Problem Definition and the related discussion to clarify vague system boundaries. The explanation now covers client roles, participation stability, and data ownership, with healthcare and IoT examples.

Comment 6:

The paper introduces the AFC framework but the introduction section has many issues. The structure is unclear, and there is no roadmap to guide the reader. Subheadings appear suddenly and disrupt the flow, while too much technical detail is placed too early in the paper. The problem statement is scattered and unclear, and the AFC framework is introduced too late, making the contribution less visible.

Response:

I reorganized the Introduction. A clear roadmap is now provided. Technical details were shifted to the Methodology, while the Problem Statement and Objectives are presented in a structured manner.

Comment 7:

The motivation for AFC is also not strong enough, as the introduction does not show why AFC is needed compared to existing research. Transitions between sections are abrupt, and the objectives and contributions are not clearly linked to the challenges. The introduction should be reorganized so that the challenges flow smoothly into the problem statement, followed by the proposed AFC framework. The objectives should be listed clearly, and the novelty of AFC should be emphasized.

Response:

The Introduction has been rewritten to strengthen motivation. Section flow was improved so that challenges lead directly into the problem statement, objectives, and then the AFC framework. The novelty of AFC is emphasized with explicit comparisons to prior work.

Comment 8:

There are also minor issues such as grammar errors, awkward sentences, and too much technical detail in the introduction that should be moved to the related work section. The paper also does not discuss any limitations of AFC, which would help present a more balanced perspective. Overall, the paper has promise, but the introduction needs major revision to improve clarity, structure, and motivation.

Response:

I corrected grammar and sentence structure issues, moved technical details from the Introduction to Related Work, and added a Limitations and Future Work section.

Comment 9:

Figure 2 does not clearly show the concept of adaptive federated clustering and selection. The labeling is confusing and the arrows are repetitive. The explanation in the methodology section also does not clarify the figure well. Both the figure and the related text should be revised for better understanding.

Response:

Figure 2 has been redesigned with clearer labels and simplified arrows. The explanation in the Methodology Section was rewritten to align directly with the figure.

Comment 10:

The dataset section is weak because it does not explain why the chosen dataset, MIMIC-III, was selected. The connection between the dataset and the main concept of the paper is not clear. The authors should explain how the dataset relates to the study and why it is suitable.

Response:

I expanded the Dataset Section to justify the selection of MIMIC-III. The description now explains its relevance to healthcare, decentralized silos, and privacy-preserving FL scenarios.

Summary:

I sincerely thank the reviewers for their detailed and constructive feedback. All comments were carefully addressed through major revisions, including structural improvements, grammar corrections, expanded experimental evaluation, clarification of vague concepts, and stronger motivation. I believe the revised manuscript is now significantly improved in clarity, technical depth, and alignment with the journal’s standards.

---

## [Decision Letter · Decision Letter 1]

5 Nov 2025

Adaptive Federated Clustering for Uncertainty-Aware Learning on Decentralized Big Data Platforms

PONE-D-25-43938R1

Dear Dr. Alhazmi,

We’re pleased to inform you that your manuscript has been judged scientifically suitable for publication and will be formally accepted for publication once it meets all outstanding technical requirements.

Kind regards,

Abdul Ahad, PhD

Academic Editor

PLOS ONE

Reviewers' comments:

Reviewer's Responses to Questions

**Comments to the Author**

Reviewer #1: All comments have been addressed

Reviewer #2: All comments have been addressed

2. Is the manuscript technically sound, and do the data support the conclusions?

Reviewer #1: Yes

Reviewer #2: Yes

3. Has the statistical analysis been performed appropriately and rigorously?

Reviewer #1: Yes

Reviewer #2: Yes

4. Have the authors made all data underlying the findings in their manuscript fully available?

Reviewer #1: Yes

Reviewer #2: No

5. Is the manuscript presented in an intelligible fashion and written in standard English?

Reviewer #1: Yes

Reviewer #2: Yes

Reviewer #1: The authors have incorporated all the comments and no further feedback is required. Authors are always welcomed to further proofread their manuscript before publication.

Reviewer #2: (No Response)

**Do you want your identity to be public for this peer review?** For information about this choice, including consent withdrawal, please see our Privacy Policy

Reviewer #1: **Yes**

Reviewer #2: No

---

## [Editor Report · Acceptance letter]

PONE-D-25-43938R1

PLOS ONE

Dear Dr. Alhazmi,

I'm pleased to inform you that your manuscript has been deemed suitable for publication in PLOS ONE. Congratulations! Your manuscript is now being handed over to our production team.

Kind regards,

on behalf of

Dr. Abdul Ahad

Academic Editor

PLOS ONE